# Studies of the Application of Electrically Conductive Composite Copper Films to Cotton Fabrics

Ramshad Abzhalov [1], Malik Sataev [1], Shaizada Koshkarbayeva [1], Guzaliya Sagitova [2], Bakyt Smailov [3], Abdugani Azimov [3], Bagdagul Serikbaeva [1], Olga Kolesnikova [4,*], Roman Fediuk [5,6,*] and Mugahed Amran [7,8]

1. Department of "Chemical Technology of Inorganic Substance" M. Auezov, South Kazakhstan University, Shymkent 160012, Kazakhstan
2. Department of "Oil Refining and Petrochemistry" M. Auezov, South Kazakhstan University, Shymkent 160012, Kazakhstan
3. Department of "Standardization and Certification" M. Auezov, South Kazakhstan University, Shymkent 160012, Kazakhstan
4. Department of Science, Production and Innovation M. Auezov, South Kazakhstan University, Shymkent 160012, Kazakhstan
5. Polytechnic Institute, Far Eastern Federal University, Vladivostok 690922, Russia
6. Department of Building Materials Science, Products and Structures, Belgorod State Technological University Named after V.G. Shoukhov, 308012 Belgorod, Russia
7. Department of Civil Engineering, College of Engineering, Prince Sattam Bin Abdulaziz University, Alkharj 16273, Saudi Arabia
8. Department of Civil Engineering, Faculty of Engineering and IT, Amran University, Amran 9677, Yemen
* Correspondence: ogkolesnikova@yandex.kz (O.K.); fedyuk.rs@dvfu.ru (R.F.); Tel.: +7-705-256-6897 (R.F.)

**Abstract:** This paper presents a technology for applying copper and silver films to cotton fabrics by combining photochemical and chemical methods for the reduction of the compounds of these metals. The resulting metal-containing films have inherent electrical conductivity of metals. All the main processes described in the work were carried out by means of the compounds being sorbed by the surface of the fabric when they were wetted in appropriate solutions. The aim of the work was to study the application of electrically conductive composite copper films on cotton fabrics. The tasks to achieve this aim were to perform scanning electron microscopy, energy dispersive spectroscopy and X-ray diffraction analysis to confirm that as a result of the experiment, CuCl with a semiconductor ability was formed on the surface of the sample. The driving force behind the photochemical reduction of copper and silver halides on cotton surfaces is that, as a result of the photooxidation of cellulose molecules in the fabric, copper monochloride is first formed on the cotton surface. Subsequently, the process of obtaining silver particles based on semiconductor silver chloride obtained as a result of the transformation of copper monochloride was carried out. The physicochemical and photochemical processes leading to the formation of monovalent copper chloride, which provides sufficient adhesion to the substrate, are considered. It is shown that in this case, the oxidation of monovalent copper also occurs with the formation of soluble salts that are easily removed by washing. Since the proposed technology does not require special equipment, and the chemical reagents used are not scarce, it can be used to apply bactericidal silver films to various household items and medical applications in ordinary laundries or at home. This article examines an affordable and simple technology for producing metal films on a cotton surface due to the presence of disadvantages (time duration, high temperature, scarce reagents, special installations, etc.) of a number of well-known methods in the production of chemical coatings.

**Keywords:** cotton fabric; photochemical reduction; composite; copper monochloride; copper; silver; conductive thin films

## 1. Introduction

At present, interest has increased in dielectric materials containing copper and silver films due to their protective properties against bacteria, viruses and various radiations [1–3]. The bactericidal and antimicrobial activities of silver have been known for a long time. Developments devoted to the deposition of silver films on medical materials have become widespread, such as implants, drug delivery systems, antibacterial coatings for biomedical devices and antimicrobial packaging [4–6]. Such materials are also used in household products: curtains, napkins, bandages, bactericidal inserts in various appliances (refrigerators, fans, air conditioners) and clothing items (socks, insoles, etc.) [5,7,8].

To apply silver to fabrics, physical and chemical methods are used. Of the physical methods, one can note the use of magnetron sputtering in a vacuum chamber of metal particles (including silver) with subsequent deposition on the surface of the textile materials [9–11]. The method is based on the use of an anomalous discharge of positively charged ions formed in a discharge in inert gases. Here, positively charged ions in the process act on the surface of the cathode and release metal particles from it, which are subsequently deposited onto the surface of the material being processed in the form of a thin film. An increase in the adhesion ability of the obtained film is provided by the high kinetic energy of the particles leaving the cathode surface.

To obtain antimicrobial silk fibers modified with silver nanoparticles, it is proposed to use $\gamma$-radiation [12]. Silk fibers treated with $AgNO_3$ solution and $\gamma$-irradiation showed 96% antimicrobial activity, which after 10 washing cycles, was maintained above 85%.

Physical methods include the technology of obtaining medical dressings, in which silver particles deposited as a result of a chemical reaction are physically dispersed between fibrous tissues [13]. The main obstacles in the use of physical methods are the use of special equipment, the need for preliminary preparation of metal nanoparticles, the difficulty of metallizing internal surfaces, and the difficulty of controlling the coating thickness.

Therefore, chemical methods are often used to apply silver-containing films to fabrics. To obtain a bactericidal textile material, silver is first obtained from an aqueous solution of silver nitrate with chemical reducing agents (ascorbic acid, glucose, hydrazine or hydrazine hydrate) and then applied to the textile material [14].

In [15], materials based on tannin were used to obtain bactericidal tissues by the reduction of silver nitrate. In the process, textile materials are processed in a heated solution of tannins. Then, silver is applied to their surface by reduction. This method goes through multi-stage operations, so it takes a lot of time.

To obtain bactericidal films of particles of Cu, Ag and Au, photochemical methods were also used [16,17]. A known method of photochemical reduction of compounds of transition metals on the surface of fabric samples allows obtaining metal nanoparticles that act as sorption centers, in the form of a stable dispersion. Photochemical synthesis of copper and silver films on the surface of tissue was carried out in previous studies.

In general, chemical methods are distinguished by multi-stage technology. Sufficient adhesion strength of metallic silver particles to a fabric material is not always ensured; they require the consumption of chemical compounds. The review shows that due to the wide variety of textile materials in terms of physicochemical properties and composition, in some cases, the application of existing methods can be difficult, so the creation of new methods for the introduction of silver is relevant.

Currently, there are many known methods for obtaining multifunctional metal films on the surface of fabric materials in the production of packaging [10–16]. However, a number of the above methods have their shortcomings, such as the length of time and the use of special installations and expensive and inaccessible chemical reducing agents. The study of an affordable and simple technology for producing a metal film on this cotton surface is the main advantage of the present work.

The aim of this work was to study the application of electrically conductive composite copper films on cotton fabrics. The tasks to achieve this aim were to perform scanning electron microscopy, energy dispersive spectroscopy and X-ray diffraction analysis to

confirm that as a result of the experiment, CuCl with semiconductor ability was formed on the surface of the sample.

## 2. Materials and Methods

For the study, cotton fabric (article AA010278), widely used in everyday life and medicine, was used. Cotton fabric is a natural and environmentally friendly material. It absorbs moisture well due to adhesion, and passes air because the density of the cotton fabric is 60 g/cm$^2$. The fabric consists of 100% cotton (98% cellulose) and does not stretch during washing. Its elemental composition is C—47.86 wt % and 55.01 atomic %; O—52.14 wt % and 44.99 atomic %. After washing, the samples were moistened by immersion in a 50 g/L CuCl$_2$ solution for 5–10 min. Then, the samples were placed on a glass or polymer surface and leveled with a glass rod. The amount of CuCl$_2$ solution absorbed by the tissue was 1 mL/dm$^2$. After that, the samples were dried under the influence of sunlight (40 min, 1000–1200 W/m$^2$). Sunlight is often referred to as visible solar rays that have a wavelength of 400 to 700 nm. The energy flux density of solar radiation reaching the Earth's surface reaches up to 1.4 kW/m$^2$. Light waves can also penetrate solid bodies, but their intensity decreases. In this case, an important characteristic of the rays is the density of the solar radiation flux [18–21]. To determine this value, an SM 206-SOLAR solar flux meter SANPOMETER (Shenzhen, China) was used [19,20].

For experiments, the samples were placed perpendicular to the sun's rays and left open until completely dry. The color of the sample changed from green (the color of CuCl$_2$ solutions) to black. Black color is typical for fine particles of metals formed during chemical reduction [22–24].

In addition, it was found that the intensity of the black color of the film depends on the concentration of the initial CuCl$_2$ solution in which the fabric was processed. Therefore, the degree of blackening of the film can be used as an indicator characterizing the content of reduced metal particles. The quantitative characteristics of the intensity of the black films of the samples can be determined using a computer by finding the degree of brightness of the pattern in the sample window. To this end, photographs of samples obtained at different stages of the process were placed on white paper, and brightness was added to each sample until the sample image completely disappeared. This was the additional blackening of the film in the brightness sample (as a percentage). For example, the degree of black in the computer's color palette was 100%.

Figure 1 shows fabric samples before (a) and after (b) the application of a photochemical copper film. Measurement of the degree of blackening by the above method showed that for the original fabric, this value was 22%, and after applying the film, this value was 75% (degree of blackening of the film, Figure 1b).

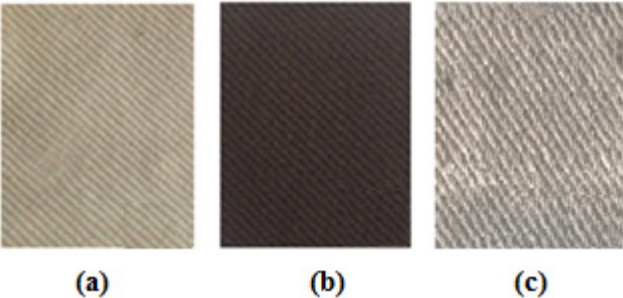

**Figure 1.** Photo of tissue before (**a**) and after (**b**) deposition of a photochemical copper film and after (**c**) deposition of a silver film.

To remove the reaction by-product (HCl) and excess CuCl$_2$ after photochemical treatment, the samples were washed with a sufficient amount of distilled water. The samples

were then dried at room temperature for 25–30 min. A slightly moistened cloth contained only copper chloride (1).

$$2CuCl_2 + H_2O + R\text{-OH} = 2CuCl + 2HCl + R\text{-OOH} \tag{1}$$

where R is an elementary unit of cellulose.

After the formation of a semiconductor layer under the action of solar photons, photons give additional energy, and excited electrons acquire the ability to oxidize, which is necessary to reduce monovalent copper (2) [25].

$$CuCl + e = Cu + Cl^- \tag{2}$$

After this reaction, vacancies remain in the semiconductor, attracting electrons from reducing agents with a sufficiently high positive potential, and again forming $CuCl_2$ (3).

$$CuCl + Cl^- - e = CuCl_2 \tag{3}$$

The electromotive force (EMF) of these reactions is 0.401 V. Therefore, the additional energy received from the sun's rays must provide an actual voltage in the system that exceeds this value. In this case, a photochemical reaction takes place.

$$2CuCl \xrightarrow{h\gamma} Cu + CuCl_2 \tag{4}$$

In addition, $CuCl_2$ crystallizes and loses activity when the surface shell dries out, which is an additional factor facilitating reaction (4).

Thus, under the influence of sunlight, films consisting of particles of copper and $CuCl_2$ are formed on the surface of cotton fabric. This system is only stable in an anhydrous environment because the thermodynamic possible reverse reaction occurs in the presence of water.

$$CuCl_2 + Cu \rightarrow 2CuCl \tag{5}$$

This reduces the possibility of further processing the fabric with aqueous solutions. The formation of $CuCl_2$ is associated with the formation of reaction (3); therefore, it is necessary to add a reagent to the reaction medium that has a greater donor ability than copper monochloride. Our research experiments have shown that ascorbic acid, which is oxidized to dehydroascorbic acid, belongs to such a reducing agent.

With an increase in pH from 0 to 7, the redox potential of the ascorbic acid–dehydroascorbic system changes from −0.329 V to −0.057 V. In addition, this system is electrochemically inert, and for the manifestation of redox properties, it is necessary to add "electrode catalysts". Indeed, the experiments showed that no traces of chemical changes were found on the fabric moistened with a solution of 50 g/L $CuCl_2$ and 40 g/L A-$(OH)_2$ and dried in a dark place. At the same time, these samples, dried under the influence of sunlight, were covered with a dark shell, which is characteristic of dispersed metal particles. In addition, if individual places of the sample are protected from sunlight by polymer washers, then no changes will occur in these places. Therefore, under the influence of sunlight, ascorbic acid is activated ((6)–(7)) and promotes the passage of CuCl → Cu. In this case, the following reaction proceeds instead of reaction (3).

$$A\text{-}(OH)_2 - e \rightarrow A\text{-OOH} \tag{6}$$

where A is the unchanged part of ascorbic acid.

The overall photochemical reaction will look like

$$2CuCl + A - (OH_2) \xrightarrow{h\gamma} Cu + HCl + A - OOH \tag{7}$$

A-(OH)$_2$ and A-OH are easily removed during the washing process. Therefore, A-(OH)$_2$ contributes to the preservation of a film with elemental copper particles on the surface of the tissue, preventing the formation of CuCl$_2$. This film can be used as a chemical silver coating activator.

After the formation of monovalent copper chloride, the samples were moistened by immersion in a solution of 20 g/L AgNO$_3$ for 5–10 min. Then, the samples were dried under the influence of sunlight (500–600 W/m$^2$). Next, the samples were moistened with a solution of 40 g/L A-(OH)$_2$. During the transformation of copper monochloride (Figure 1c) into silver, the following reactions occur ((8)–(9)):

$$AgNO_3 + HCl \rightarrow AgCl + HNO_3 \qquad (8)$$

$$2AgNO_3 + C_6H_8O_6 + H_2O \rightarrow C_6H_8O_7 + 2Ag + 2HNO_3 \qquad (9)$$

As a result of all these processes, the surface layer of the fabric will contain only a single-chloride copper that is adequately adherent to the base. In this case, the fabric acquires a uniform, monophonic light green color, which indicates the formation of copper chloride associates with macromolecules of cellulose. The formation of these associates is possible and is associated with high adhesion of the films. The last impregnation of the tissue with a solution of silver nitrate leads to the course of the exchange reaction involving copper chloride and the formation of a photosensitive layer of silver chloride on it under the action of a solar photon.

These data allow us to propose the following scheme of photochemical processes occurring under the influence of electromagnetic waves of solar rays (Figure 2).

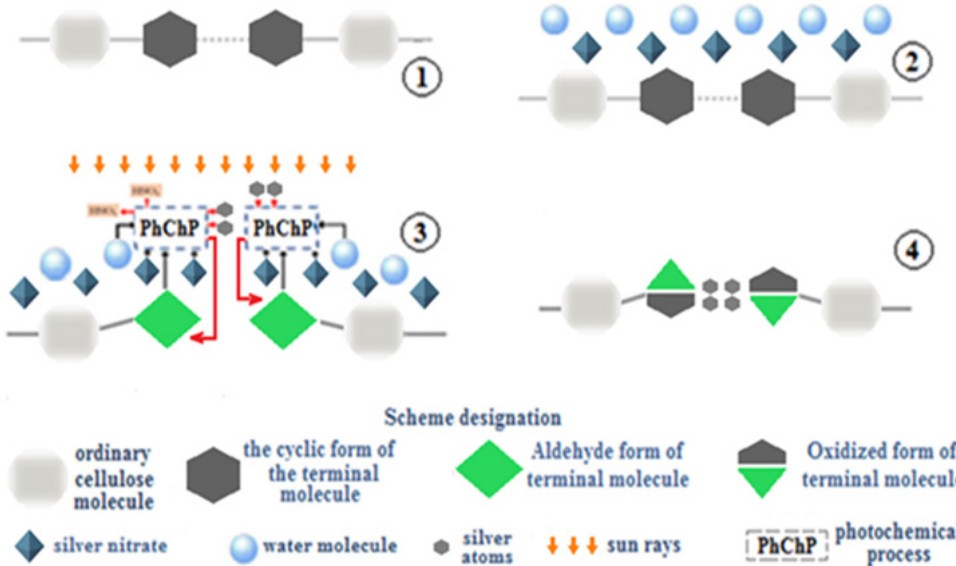

**Figure 2.** General scheme of the processes occurring with the participation of the end groups of cellulose under the influence of electromagnetic waves of sunlight. Designations: (**1**) main molecules of the initial end group; (**2**) end group molecules containing a sorption layer of silver nitrate solution; (**3**) processes occurring under the influence of electromagnetic waves of sunlight; (**4**) end group molecules and silver atoms after the photochemical process.

Silver particles formed by the photochemical reaction (13) are surrounded by many wavelets [26] (the term "wavelet" in translation from English means "small (short) wave"). These areas are reflected by light in the form of black shadows between individual metal particles. As a result, the resulting silver layers are black. At the same time, wavelets do not allow individual dispersed particles to merge into larger ones. These films are not continuous and electrically conductive, since between individual particles, it is impossible

for electrons to flow from one particle to another. These particles are usually only activators of the chemical metallization process.

The study of the composition and structure of the films at individual stages of the process was carried out using an ISM-6490-LV scanning electron microscope (JEOL, Kyoto, Japan) and a D8 Advance X-ray diffractometer (Bruker). These devices make it possible to determine the electronic image of tens of nanometer particles, the elemental composition, the phase composition of the film and the percentage of elements in the film surface layers.

To determine the electrical conductivity of the obtained films, a DT-830B resistance tester was used.

### 3. Results and Discussion

Figure 3 shows the results of scanning electron microscopy, confirming that, as a result of the experiment, CuCl, which has semiconductor ability, is formed on the surface of the sample in accordance with reaction (1). By calculating the EDS results by percentage, one can verify the formation of monovalent copper chloride.

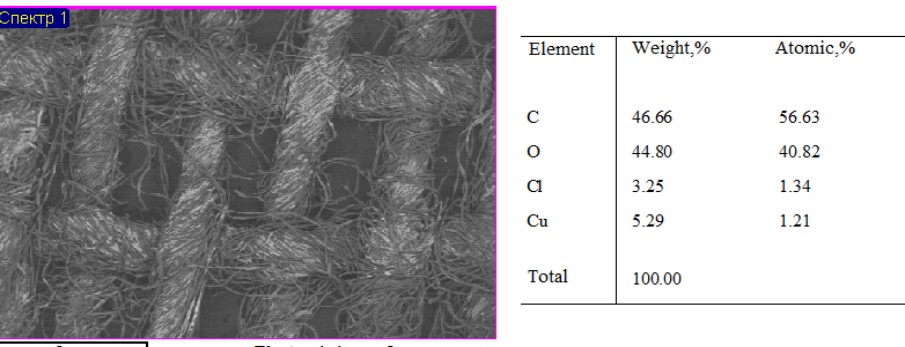

**Figure 3.** SEM images and elemental composition of a copper (I) chloride film on cotton fabric.

Figure 4 is an X-ray pattern showing that the film formed on the cotton surface is copper monochloride. Here, as compounds, $(C_6H_{10}O_5)_n$—79.1% and CuCl—20.9% are formed on the cotton surface.

The semiconductor film of copper (I) chloride obtained on a cotton surface was used as an activator for further metallization of the sample. It can be said that the agent acts as an activator for planting a silver film on the surface of the sample. By transforming the CuCl film formed on the surface of the sample according to reactions (8) and (9), a silver film is obtained (Figure 1c).

To determine the suitability of the obtained films for electroplating, the electrical conductivity of silver films was measured.

Under the action of a silver nitrate solution, catalytic centers consisting of silver particles are formed ((10)–(11)), and copper and silver are exchanged in the activator particles.

$$2AgNO_3 + Cu = 2Ag + Cu(NO_3)_2 \tag{10}$$

$$2AgNO_3 + CuCl_2 = 2AgCl + Cu(NO_3)_2 \tag{11}$$

In this case, silver chloride is a light-sensitive binary semiconductor on which, under the influence of photons of sunlight, a reduction reaction occurs (12):

$$AgCl \overset{h\gamma}{\to} Ag + Cl^- \tag{12}$$

Figure 5 shows the elemental composition and SEM images of the silver film obtained on the sample surface. There is evidence to conclude that, according to the results of scanning electron microscopy, an Ag film (26.48%) was obtained on the sample surface.

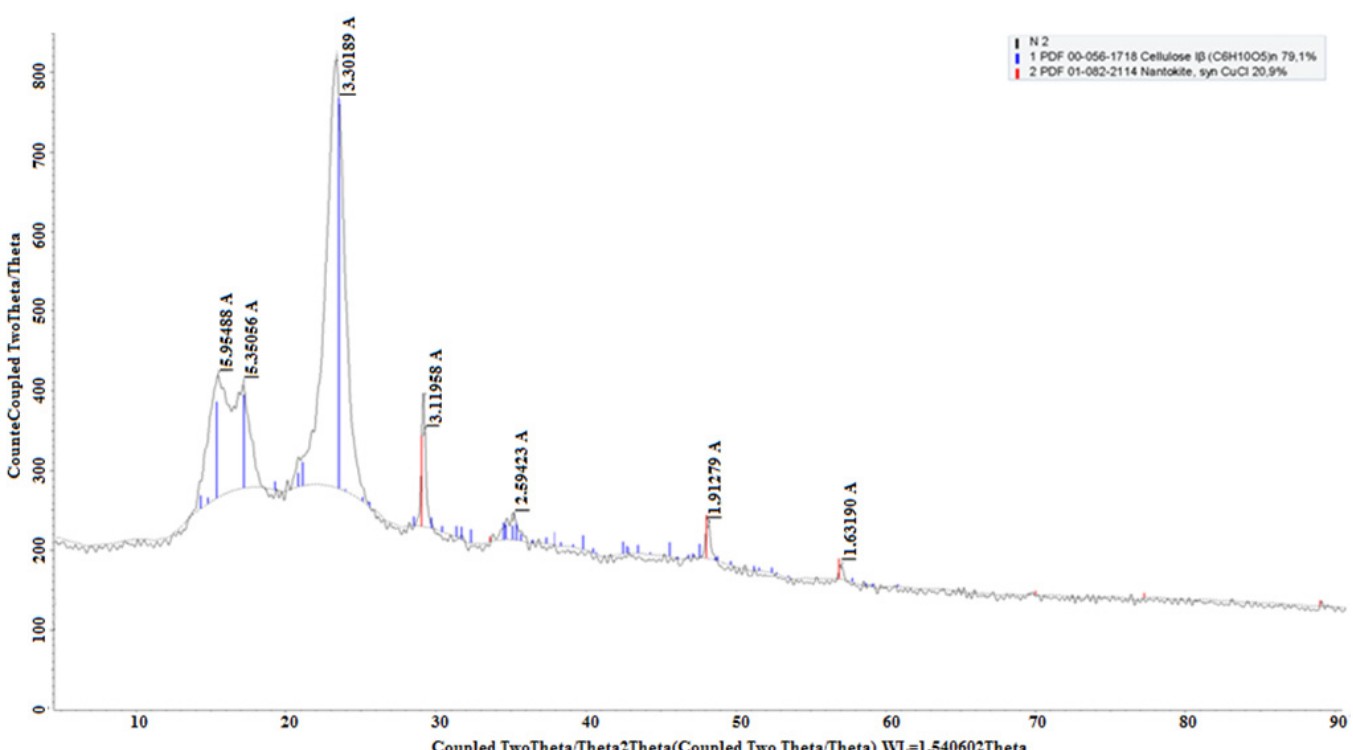

**Figure 4.** X-ray patterns of copper (I) chloride film on cotton fabric.

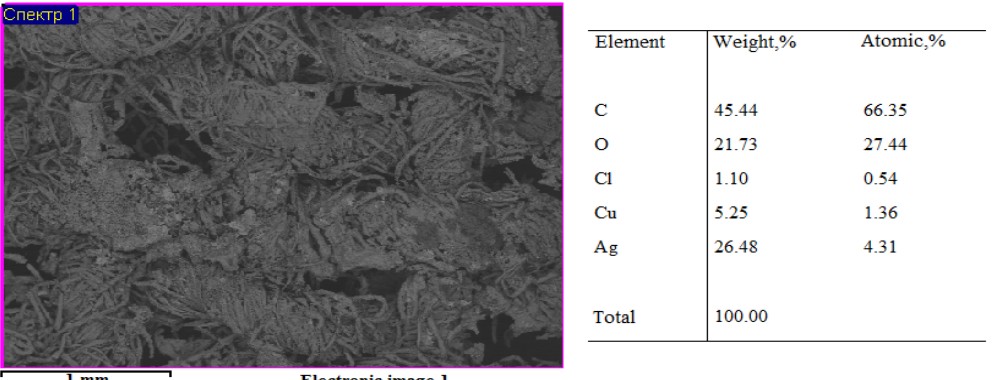

| Element | Weight,% | Atomic,% |
|---------|----------|----------|
| C | 45.44 | 66.35 |
| O | 21.73 | 27.44 |
| Cl | 1.10 | 0.54 |
| Cu | 5.25 | 1.36 |
| Ag | 26.48 | 4.31 |
| Total | 100.00 | |

**Figure 5.** SEM images and elemental composition of the silver film on cotton fabric.

According to Figure 5, the structure of the resulting film on the surface of cotton fabric materials consists of silver, copper, and chloride ions. Moreover, based on the chlorine content, copper and silver are mainly in atomic form.

Figure 6 shows the composition of the surface layer formed after the photochemical process. A silver film X-ray pattern obtained on a cotton surface shows the quantity of reduced silver present on the surface of the cellulose. Moreover, copper can be partially found in the form of initial compounds. The components of the surface layer of the sample are Ag (65.0%), $(C_6H_{10}O_5)n$ 24.8% and $Cu_2(NO_3)(OH)_3$ 10.2%.

Thus, the formation of silver films on the surface of the fabric occurs due to photochemical processes (12) (which do not lead to the formation of electrically conductive silver films) and chemical processes ((9), (10)) (leading to the formation of electrically conductive silver films). By increasing the share of the second type of processes, it is possible to achieve the production of films with sufficient electrical conductivity. The resistance of the surface film at a distance of 1 cm was $0.07 \times 10^{-2}$ Ω.

All the main processes cited in this work were carried out by means of the compounds being sorbed by the tissue surface when they were wetted in appropriate solutions. By-products were easily removed by washing the fabric. This approach eliminates the use of large volumes of chemical reduction solutions and, accordingly, there is no need for frequent adjustments during the process. This makes it convenient to use this technology in various medical institutions.

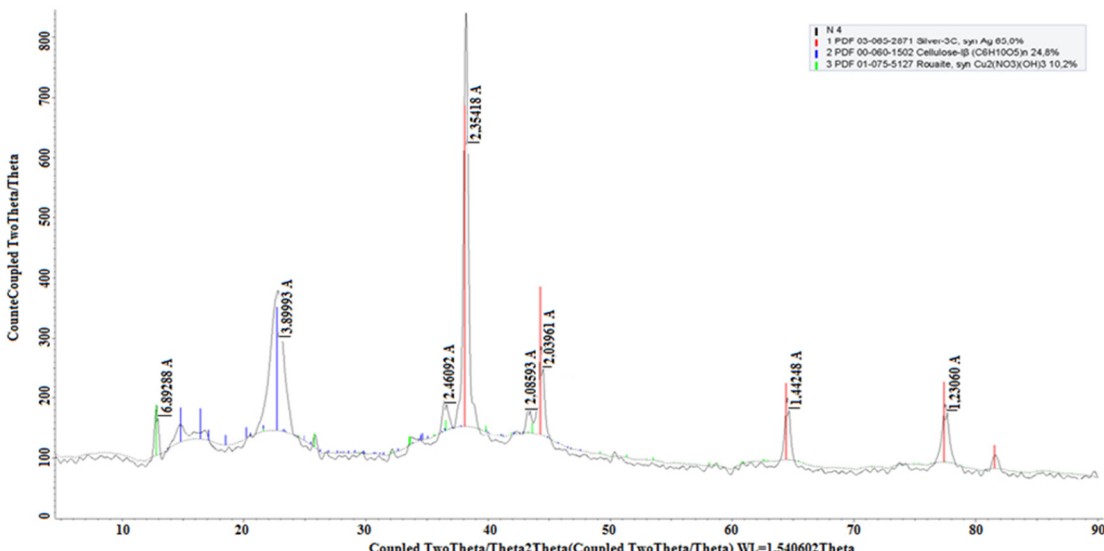

**Figure 6.** X-ray patterns of the silver film on cotton fabric.

## 4. Conclusions

We conducted studies of the application of electrically conductive composite copper films to cotton fabrics. The tasks to achieve this aim were to perform scanning electron microscopy, energy dispersive spectroscopy and X-ray diffraction analysis to confirm that as a result of the experiment, CuCl with semiconductor ability was formed on the surface of the sample. The following results were obtained, showing the scientific novelty and practical significance of the research.

By the action of electromagnetic waves of sunlight on the surface of a cotton fabric moistened with a solution containing 50 g/L of $CuCl_2$, 20 g/L of $AgNO_3$ and 40 g/L of ascorbic acid, electrically conductive films consisting of copper and silver were obtained. Studies have shown that this is the result of photochemical and chemical processes occurring in the sorption layer on the surface of cotton fabric under the influence of sunlight. As a result of the reaction of cellulose with copper (II) chloride, monovalent copper chloride is initially formed. This copper compound is a semiconductor, and under the action of photons of sunlight, a photochemical reaction of the formation of elemental copper occurs, which serves as a catalyst for further chemical silvering with ascorbic acid. The resulting silver layer had a sufficiently low electrical resistance of $0.07 \times 10^{-2}$ Ω. This allows the electroplating process to be used for different metal coatings on fabrics.

**Author Contributions:** Conceptualization, M.S.; Data curation, R.A.; Formal analysis, S.K.; Funding acquisition, R.F.; Methodology, G.S. and B.S. (Bakyt Smailov); Project administration, A.A.; Resources, B.S. (Bagdagul Serikbaeva); Software, O.K.; Visualization, M.A.; Writing—original draft, R.A., M.S., S.K., G.S., B.S. (Bakyt Smailov), A.A., B.S. (Bagdagul Serikbaeva), O.K., R.F. and M.A. All authors have read and agreed to the published version of the manuscript.

**Funding:** The study was supported by the RSF grant no. 22-19-20115, https://rscf.ru/project/22-19-20115/ and the Government of the Belgorod Region, Agreement no. 3 of 24 March 2022.

**Data Availability Statement:** Not applicable.

**Conflicts of Interest:** The funders had no role in the design of the study; in the collection, analyses, or interpretation of data; in the writing of the manuscript; or in the decision to publish the results.

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
