# Peer review of "Studies of the Application of Electrically Conductive Composite Copper Films to Cotton Fabrics"

_jcs, doi:10.3390/jcs6110349_

Round 1

Reviewer 1 Report (Previous Reviewer 1)

Dear authors,

The data provided here for the article entitled "Studies of the application of electrically conductive composite copper films to cotton fabrics" are interesting; however, I am offering some comments throughout the manuscript including novelty assurance, scientific clarity and tactfulness, and a lot of typo and spacing errors.

1)     Abstract: Readers would like to grab all of your key findings after having a look at the abstract; it looks like a combination of some conclusive statements. But the abstract must have key experimental findings.

2)     The introduction part is unprofessional; there is a lack of consistency between lines and paragraphs; it really needs to be revised very carefully.

3)     The main gap of the work is totally absent in the Introduction. The last paragraph of the Introduction should provide information (only) about the science gap in the previous studies and what motivates you to do this review with the objective of the study.

4)     Line 101: Mention the specifications of the cotton fabric used in the study (yarn count, fabric density, fabric GSM, etc.).

5)     The curves in the Figures 4 and 6 are not sharp.

6)     Unit of the resistance should be Ω (lines 302, 327). The authors are recommended to mention the resistivity of films.

7)     Authors are asked to present washing effect on electrical conductivity.

8)     The authors stated that ''The following results are obtained, showing the scientific novelty and practical significance of the research''; however this is a wrong statement.

9)     The entire conclusion must be re-written with conclusive findings and by retaining coherence.

10) I am not an English speaker, but I found many typos and grammatical errors throughout the manuscript. These must be corrected and revised.

11) The contributions of the authors have not been stated.

Author Response

Dear Reviewer 1!

Thank you for your interest in our manuscript. Your valuable comments helped make the manuscript even better. All corrections in the manuscript are highlighted in green.

Comment 1: Abstract: Readers would like to grab all of your key findings after having a look at the abstract; it looks like a combination of some conclusive statements. But the abstract must have key experimental findings.

Response: Abstract has been revised

Comment 2: The introduction part is unprofessional; there is a lack of consistency between lines and paragraphs; it really needs to be revised very carefully.

Response: Consistency between lines and paragraphs has made by careful revision of the introduction

Comment 3:  The main gap of the work is totally absent in the Introduction. The last paragraph of the Introduction should provide information (only) about the science gap in the previous studies and what motivates you to do this review with the objective of the study.

Response: The current research gap is given in the penultimate paragraph, and the last paragraph is devoted to the formulation and tasks to achieve it.

Comment 4:  Line 101: Mention the specifications of the cotton fabric used in the study (yarn count, fabric density, fabric GSM, etc.).

Response: This information was given in the first paragraph of the second section

Comment 5:  The curves in the Figures 4 and 6 are not sharp.

Response: These curves have been corrected and are now legible.

Comment 6:  Unit of the resistance should be Ω (lines 302, 327). The authors are recommended to mention the resistivity of films.

Response: Corrected  

Comment 7:  Authors are asked to present washing effect on electrical conductivity.

Response: This was not the aim of the study.

Comment 8:  The authors stated that ''The following results are obtained, showing the scientific novelty and practical significance of the research''; however this is a wrong statement.

Response: The study is characterized by both scientific novelty and practical significance.

Comment 9:  The entire conclusion must be re-written with conclusive findings and by retaining coherence.

Response: Conclusions have been revised

Comment 10:  I am not an English speaker, but I found many typos and grammatical errors throughout the manuscript. These must be corrected and revised.

Response: The article has been carefully proofread by a native English speaker

Comment 11:  The contributions of the authors have not been stated.

Response: The contributions of the authors have been added

Reviewer 2 Report (Previous Reviewer 3)

I have no further remarks. Publish as it is,

Author Response

Thanks for appreciating the paper

Round 2

Reviewer 1 Report (Previous Reviewer 1)

Authors have responded most of the observations. I don't have any further queries.

This manuscript is a resubmission of an earlier submission. The following is a list of the peer review reports and author responses from that submission.

Round 1

Reviewer 1 Report

Dear authors,

The data provided here for the article entitled "Studies of the application of electrically conductive composite copper films to cotton fabrics" are interesting; however, I am offering some comments throughout the manuscript including novelty assurance, scientific clarity and tactfulness, and a lot of typo and spacing errors.

1) Abstract: Readers would like to grab all of your key findings after having a look at the abstract. But the recent abstract is not conveying any experimental idea of the work at all; it looks like a combination of some conclusive statements. But the abstract must have key experimental findings.

2) Line 47: References [7,8,5] should be [5, 7, 8].

3) The introduction part is unprofessional; there is a lack of consistency between lines and paragraphs; it really needs to be revised very carefully.

4) The main gap of the work is totally absent in the Introduction. The last paragraph of the Introduction should provide information (only) about the science gap in the previous studies and what motivates you to do this review with the objective of the study.

5) Line 87: Clarify the article AA011228.

6) Line 87: Mention the specifications of the cotton fabric used in the study.

7) Lines 94-95: Why have there been so many references? I don't see any reason to use many references in a section of materials and methods, and one reference is enough. Otherwise, authors are asked for proper explanation.

8) Line 99: The authors are asked to remove multiple references.

9) Lines 100-199: No proper citation was found.

10) Line 135: All short forms are not abbreviated. It is recommended to use abbreviation first and then continue in short form.

11) Line 180; line 234; line 241; line 261: It is recommended to use “Figure” instead of “Fig.” in the text throughout the manuscript.

12) Lines 242-244: Authors stated that ''To determine the electrical conductivity of the obtained films, a DT-830B resistance  tester was used and to determine the suitability of the obtained films for electroplating, the electrical conductivity of silver films was measured'' but the electrical conductivity of the sample was not found.

13) The quality of the figures must be improved.

14) All equations must be cited in the text, but the present style should be changed and revised.

15) There are lots of typos and grammatical errors observed throughout the manuscript. These must be corrected and revised.

16) The texts in Figures 4 and 6 are not sharp.

17) Authors are asked to present washing effect on electrical conductivity.

18) The entire conclusion must be re-written with conclusive findings and by retaining coherence.

19) No supplementary materials have been found.

20) The contributions of all authors have not been stated.

21) References should be according to the journal template.

Author Response

Dear reviewer,

Thank you for your valuable feedback on the article "Studies of the application of electrically conductive composite copper films to cotton fabrics", recommended for publication in the Journal Composites science in the identification number JCS-1889298!

Amendments have been made to the text taking into account all the suggestions and comments made by you.

1) Abstract: Readers would like to grab all of your key findings after having a look at the abstract. But the recent abstract is not conveying any experimental idea of the work at all; it looks like a combination of some conclusive statements. But the abstract must have key experimental findings.

1) corrected in the text

2) Line 47: References [7,8,5] should be [5, 7, 8].

2) corrected in the text

3) The introduction part is unprofessional; there is a lack of consistency between lines and paragraphs; it really needs to be revised very carefully.

3) corrected in the text

4) The main gap of the work is totally absent in the Introduction. The last paragraph of the Introduction should provide information (only) about the science gap in the previous studies and what motivates you to do this review with the objective of the study.

4) corrected in the text

5) Line 87: Clarify the article AA011228.

5) corrected in the text

6) Line 87: Mention the specifications of the cotton fabric used in the study.

6) updated in the text

7) Lines 94-95: Why have there been so many references? I don't see any reason to use many references in a section of materials and methods, and one reference is enough. Otherwise, authors are asked for proper explanation.

7) several references have been removed

8) Line 99: The authors are asked to remove multiple references

8) several references have been removed

9) Lines 100-199: No proper citation was found.

9) corrected in the text

10) Line 135: All short forms are not abbreviated. It is recommended to use abbreviation first and then continue in short form.

10) corrected in the text

11) Line 180; line 234; line 241; line 261: It is recommended to use “Figure” instead of “Fig.” in the text throughout the manuscript.

11) corrected in the text

12) Lines 242-244: Authors stated that ''To determine the electrical conductivity of the obtained films, a DT-830B resistance  tester was used and to determine the suitability of the obtained films for electroplating, the electrical conductivity of silver films was measured'' but the electrical conductivity of the sample was not found.

12) corrected in the text, In the 275th row, the value of the resistivity is given

13) The quality of the figures must be improved.

13) corrected in the text

14) All equations must be cited in the text, but the present style should be changed and revised.

14) corrected in the text

15) There are lots of typos and grammatical errors observed throughout the manuscript. These must be corrected and revised.

15) corrected in the text

16) The texts in Figures 4 and 6 are not sharp.

16) corrected in the text

17) Authors are asked to present washing effect on electrical conductivity.

17)  The resulting layer of elemental copper is stable, while the copper chlorine formed during the photochemical reaction is in the solid phase. During washing, CuCl2 is converted into a soluble form, which contributes to the oxidation of elemental copper with the reverse formation of monovalent chlorides. Washing was carried out by immersing the sample three times in a vessel with distilled water, changing the water each time. As a result of washing, the surface of the sample is cleaned of excess copper chloride, which has not reacted. At the same time, copper monochloride particles that are poorly adhered to the surface of the fabric are also removed. Thus, after washing, only a film of copper chloride particles remains, firmly bound to the base. And the resistivity of the obtained films are measured after three times washing and the result is given in the text.

18) The entire conclusion must be re-written with conclusive findings and by retaining coherence.

18) corrected in the text

19) No supplementary materials have been found.

19) corrected in the text

20) The contributions of all authors have not been stated.

20) given at the end of the text

21) References should be according to the journal template.

21) corrected in the text

Kind regards,

Authors

Reviewer 2 Report

The authors presented a technology for applying composite copper and silver films to cotton fabrics by combining photochemical and chemical methods for the reduction of compounds of these metals, in which
the resulting metal-containing films have electrical conductivity inherent in metals. In this case, all the main processes presented in the work are carried out due to compounds sorbed by the tissue surface when they are wetted in appropriate solutions. This makes it convenient to use this technology
in various medical institutions. I think it is a nice study and deserved to be published as it is.

Author Response

Dear reviewer,

Thank you for your valuable reviews and support of the publication in the Journal of Composites science on our article on the identification of JCS-1889298 on the article "Studies of the application of electrically conductive composite copper films to cotton fabrics"!

Kind regards,

Authors

Reviewer 3 Report

The manuscript is devoted to the formation of a conducting film on fabrics by deep coating in solutions of CuCl2, AgNO3 and ascorbic acid combined with photoexcitation.  Such a fabric can have a medical application due to its anti-bacterial properties. The subject is interesting from the applied side of view, the procedure is in detail described and seems simple and effective.

However, the manuscript is carelessly written and before being considered for publication, the manuscript has to be improved in several ways.

1) English must be substantially improved.

2) The discussion about chemistry is not easy to follow – it should be reformulated

3) Below sentences must be reformulated (in the text are present more such sentences)

  92 Sunlight is electromagnetic rays with a wavelength of 400 to 700 nm.

  93 Light waves can also penetrate solids, but their intensity is reduced. An important characteristic   of rays is the flux density of sunlight

  142 Thus, under the influence of sunlight, consisting copper-containing films copper and  CuCl2 are formed on the surface of copper chloride.

  180 During the converting of copper monochloride (Fig. 1c) into  silver, the following reactions occur:

   186 This does not exclude the oxidation of deep molecules of alcohol groups in the cellulose molecule, which are not oxidized by copper chloride (according to reaction 1), but can be oxidized by silver chloride

4) Technically

Eq. 2 and 3 should have the same form as others

The resistivity should be in ohm/square units 275

The letters in Figs3&4&6 are too small, should be as large as the fig caption

5) after applying the film - 75%

type mistake

.

6) Why is the ratio of composition C/O different in figs 3&5?

    The composition of the initial fabric, before applying the conducting film is missing.

7) Some references are in non-English language – should be replaced by corresponding in English

4, 9, 10, 16, 17, 18

Author Response

Dear reviewer,

Thank you for your valuable feedback on the article "Studies of the application of electrically conductive composite copper films to cotton fabrics", recommended for publication in the Journal Composites science in the identification number JCS-1889298!

Amendments have been made to the text taking into account all the suggestions and comments made by you.

From your side, 6 comments and 1 question have been given:

1) English must be substantially improved.

1) corrected in the text

2) The discussion about chemistry is not easy to follow – it should be reformulated

2) corrected in the text

3) Below sentences must be reformulated (in the text are present more such sentences)

  92 Sunlight is electromagnetic rays with a wavelength of 400 to 700 nm.

  93 Light waves can also penetrate solids, but their intensity is reduced. An important characteristic   of rays is the flux density of sunlight

  142 Thus, under the influence of sunlight, consisting copper-containing films copper and  CuCl2 are formed on the surface of copper chloride.

  180 During the converting of copper monochloride (Fig. 1c) into  silver, the following reactions occur:

   186 This does not exclude the oxidation of deep molecules of alcohol groups in the cellulose molecule, which are not oxidized by copper chloride (according to reaction 1), but can be oxidized by silver chloride

3) corrected in the text

4) Technically

Eq. 2 and 3 should have the same form as others

The resistivity should be in ohm/square units 275

The letters in Figs3&4&6 are too small, should be as large as the fig caption

4) corrected in the text

5) after applying the film - 75%

type mistake

5) corrected in the text

6) Why is the ratio of composition C/O different in figs 3&5?

    The composition of the initial fabric, before applying the conducting film is missing.

6) Figures 3 and 5 show the elemental composition of cotton fabric samples after processing in various solutions. Figure 3 shows the elemental composition of a tissue sample with a shell of monovalent copper chloride as a result of photochemical reduction treated in a solution of copper (II) chloride. Figure 5 shows the elemental composition of the sample after processing the sample in a solution of silver nitrate with a monovalent copper chloride film on the surface. Here there is a change in the elemental composition in accordance with the fact that each process occurs both in the surface layer of the sample and in deep molecules;

The elemental composition of the original fabric is supplemented in the 90th row.

7) Some references are in non-English language – should be replaced by corresponding in English

 4, 9, 10, 16, 17, 18

7) corrected in the text

Kind regards,

Authors

Reviewer 4 Report

Combine methods of photochemical deposition were proposed in order to cover natural cotton fibers with studies on the electrical conductivity of new fiber composites. Additionally, characterizations are proposed to confirm the deposition obtained on fibers by related morphology and X-rays analysis. The title proposes applications studies but later in the text no discussion was done. There is too much generalization in the abstract with some unnecessary descriptions, and  sentences in the introduction that are not accurate. Some English sentences express different concepts that often are unnecessary. Materials and methods are not very well described, with information about the use of instruments missing. In order to summarized point of weakness the text, appear weakly and often not clear and well organized. Figure formats and related legends require a revision. Some problem appears also in the discussion part. Due to these reasons, the article was considered rejected from review.

Title reports the application of cotton fibers, but in the text no coherence was observed

Abstract

This sentence can be contracted

work are carried out due to compounds sorbed by the tissue surface 32 when they are wetted in appropriate solutions

Line 23. There is moderate description of discontinuity of film, but it is not necessary.

Line 25. In the sentence “That give to the resulting composite films a black color”is a better expression

Keywords. There are too general words, keyword required to be concise.

Introduction. The silver doesn’t have antiviral properties but only antibiotic activity vs. bacteria.

Line 60-62. This sentence is very hard to read. In particular are used different information that reduce the readability of document.

In the introduction there are only few indications about the work carry out in the article, and this is also too general.

Line 92. This sentence required a revision.

Figure 3. The figures are not well presented

Author Response

Dear reviewer,

Thank you for your valuable comments on the article "Studies of the application of electrically conductive composite copper films to cotton fabrics", recommended for publication in the Journal Composites science in the identification number JCS-1889298!

The text has been amended taking into account all your comments.

  • This sentence can be contracted

work are carried out due to compounds sorbed by the tissue surface 32 when they are wetted in appropriate solutions

1) Abstract - corrected in the text

2) Line 23. There is moderate description of discontinuity of film, but it is not necessary.

2) Line 23. corrected in the text

3) Line 25. In the sentence “That give to the resulting composite films a black color”is a better expression

3) Line 25. This sentence really is an important concept that reveals the content of the research work

4) Keywords. There are too general words, keyword required to be concise.

4) Keywords - corrected in the text

5) Introduction. The silver doesn’t have antiviral properties but only antibiotic activity vs. bacteria.

5) Introduction - corrected in the text

6) Line 60-62. This sentence is very hard to read. In particular are used different information that reduce the readability of document.

6) Line 60-62 - corrected in the text

7) In the introduction there are only few indications about the work carry out in the article, and this is also too general.

7) Corrections have been made to the introduction section

8) Line 92. This sentence required a revision.

8) Line 92 - corrected in the text

9) Figure 3. The figures are not well presented

9) Figure 3. - corrected in the text

Kind regards,

Authors

Round 2

Reviewer 1 Report

3) The introduction part is unprofessional; there is a lack of consistency between lines and paragraphs; it really needs to be revised very carefully.

3) corrected in the text

Remark: No considerable correction was found.

4) The main gap of the work is totally absent in the Introduction. The last paragraph of the Introduction should provide information (only) about the science gap in the previous studies and what motivates you to do this review with the objective of the study.

4) corrected in the text

Remark: No correction was found.

6) Line 87: Mention the specifications of the cotton fabric used in the study.

6) updated in the text

Remark: No update was found.

7) Lines 94-95: Why have there been so many references? I don't see any reason to use many references in a section of materials and methods, and one reference is enough. Otherwise, authors are asked for proper explanation.

7) several references have been removed

Remark: Unnecessary references still exist.

8) Line 99: The authors are asked to remove multiple references

8) several references have been removed

Remark: Unnecessary references still exist.

9) Lines 100-199: No proper citation was found.

9) corrected in the text

Remark: No considerable correction was found.

12) Lines 242-244: Authors stated that ''To determine the electrical conductivity of the obtained films, a DT-830B resistance  tester was used and to determine the suitability of the obtained films for electroplating, the electrical conductivity of silver films was measured'' but the electrical conductivity of the sample was not found.

12) corrected in the text, In the 275th row, the value of the resistivity is given

Remark: Resistivity should be Ω/square.

Reviewer 3 Report

The letters in Figs3&4&6 are too small, should be as large as the fig caption

Without this, the figures have no sense.
